# Feelings of Patients Admitted to the Emergency Department

**DOI:** 10.3390/healthcare13050500

**Published:** 2025-02-26

**Authors:** Claire Pilet, Florentine Tandzi-Tonleu, Emmanuel Lagarde, Cédric Gil-Jardiné, Michel Galinski, Sylviane Lafont

**Affiliations:** 1Université Lyon, Université Gustave Eiffel, Université Claude Bernard Lyon 1, Epidemiological Research and Surveillance Unit in Transport Occupation and Environment (UMRESTTE), UMR_T9405, F-69500 Bron, France; florentine.tandzi-tonleu@univ-eiffel.fr; 2INSERM 1219—“Injury Epidemiology Transport Occupation” Team, ISPED, Bordeaux Population Health Research Centre, F-33076 Bordeaux, France; emmanuel.lagarde@u-bordeaux.fr (E.L.); cedric.gil-jardine@chu-bordeaux.fr (C.G.-J.); michel.galinski@chu-bordeaux.fr (M.G.); 3Emergency Department, Bordeaux University Hospital, F-33000 Bordeaux, France

**Keywords:** feelings, pain, stress, negative emotions, positive emotions, emergency department, gender, age, admission reason

## Abstract

**Background/Objectives**: Very few studies describe the various feelings experienced in the emergency department (ED). Our study describes the pain, stress, and negative and positive emotions experienced by patients admitted to the ED in relation to age, gender, and reason for ED admission. **Methods**: Patients admitted to the ED of seven French hospitals were surveyed as part of the randomised multicentre study SOFTER IV (n = 2846). They reported the intensity of their pain on a numerical rating scale of 0 to 10, the intensity of their stress on an equivalent scale, and their emotions on a five-point rating scale using an adapted version of the Geneva Emotion Wheel proposed by Scherer, based on eight core emotions: fear, anger, regret, sadness, relief, interest, joy, and satisfaction. **Results**: Patients reported an average pain rating of 4.5 (SD = 3.0) and an average stress rating of 3.4 (SD = 3.1). Forty-six percent reported at least one strong negative emotion, and the two most frequently reported were fear and sadness. Forty-seven percent of patients described feeling at least one strong positive emotion, and the two most frequently reported were interest and relief. Pain was significantly higher among female patients under 60 admitted for injury. Stress was significantly higher among female patients under 60 admitted for illness. Emotions of negative valency were significantly higher among women admitted for injury. Emotions of positive valency were significantly higher among men over 60 admitted for illness. **Conclusions**: Experiences of pain, stress, and emotions have a strong presence in the ED. The reporting of these feelings varies depending on age, gender, and reason for ED admission.

## 1. Introduction

Many countries are experiencing overcrowding in emergency departments (EDs), a phenomenon that is negatively affecting working conditions, meaning that patients are having to wait longer before being seen. Every year, over 20 million ED admissions are recorded in mainland France [1], corresponding to an average of 3 in every 10 French people. The event of an urgent health issue and the subsequent visit to the ED are often very significant, engendering a variety of biological and psychological responses in the individual to re-establish homeostasis. In the broader context of the motivational system [2], a health emergency implies a threat to survival and induces negative feelings, triggering the body’s fight-or-flight response. However, stimuli relating to needs—and, therefore, of interest to the body—provoke positive feelings. These feelings, therefore, relate more to patients’ reactive responses, such as pain, stress, and emotions, than to their general subjective assessment of their own circumstances. These feelings experienced in the ED, particularly stress and emotions, deserve special attention, just like with pain, which is more often explored.

Pain, in its role as a warning system that protects the organism, is very often present in the ED [3,4]. The International Association for the Study of Pain defines it as “an unpleasant sensory and emotional experience associated with, or resembling that associated with, actual or potential tissue damage” [5]. It has only recently become the subject of specific monitoring and care recommendations [6,7,8]. Nevertheless, it remains poorly treated in EDs [3,9], and 22% of patients in France report experiencing severe pain upon leaving the ED [10]. If this pain persists, it represents a risk factor for the development of chronic pain [11,12], mood dysregulation associated with anxiety and/or depression [13], and even opioid abuse [14].

Other feelings are equally present in EDs, such as stress and other high-intensity emotions. Stress can be defined as a state of worry or mental tension provoked by a difficult situation. It is a natural human response that prompts us to address challenges and threats in our lives. Everyone experiences stress to some degree. The way we respond to stress, however, makes a big difference to our overall well-being [15]. When experiencing stress, the individual is likely to perceive their environment as frightening due to excessive or prolonged secretion of cortisol, provoking a physiological reaction consisting of a feeling of unease, worry, or imminent doom. In a study by Krzyszycha et al., stress is experienced by 87% of ED patients [16]. It could also increase the risk of post-traumatic stress disorder [17]. If the stress is repeated or prolonged, it activates the body’s natural defenses by inhibiting other functions, which has negative long-term effects on health [18]. It can also lead to various health issues such as addiction, violence, insomnia, cardiovascular issues, ulcers, or even cancer [19].

Emotions are defined by Scherer as “an episode of interrelated, synchronised changes in the states of all or most of the five organismic subsystems in response to the evaluation of an external or internal stimulus event as relevant to major concerns of the organism” [20]. These five subsystems are information processing, support, executive, action, and monitoring. They are based on the following major substrata: the central nervous system, the neuro-endocrine system, the autonomic nervous system, and the somatic nervous system [21]. As for pain and stress, emotions involve the organism’s physical components. A handful of studies have documented the presence of emotions in EDs [22], but none have been carried out among study populations large enough to conduct any sort of quantitative analysis. However, recent results have shown that strong negative emotions represent a risk factor for chronic illness. If these strong negative emotions are present due to poor emotional regulation, this may also have various health consequences, such as dysfunctional eating, substance use, obesity, cardiovascular disease, coronary heart disease, diabetes, and rheumatoid arthritis [23]. These three types of feelings—pain, stress, and emotions—share common mechanisms according to the biopsychosocial model [24]. Pain can be considered to be a potential stress factor [25], and the exaggerated, prolonged, or recurrent triggering of a stress reaction can lead to or exacerbate pain or disability [26,27]. Due to its relationship with emotions, pain contains an unpleasant emotional dimension. As pain is part of the defense system, the negative effect that accompanies it renders the danger aversive [2,28]. Emotions can also impact the sensation of pain. Negative emotions exacerbate the sensation of pain and the perceived intensity of acute pain and lower the pain threshold for the unpleasant stimulus perceived as painful. Positive emotions have the opposite effect in that they attenuate this perception of pain and slightly raise the pain threshold. This modulation is even more pronounced when experiencing intense emotion [29]. As for the link between emotions and stress, the ability to regulate one’s emotions can lower the effects of stress in certain contexts [30]. These different feelings seem to interact with one another and deserve to be better understood in an emergency context.

It is likely that these feelings may be affected by personal factors, such as sociodemographic characteristics and the reason for ED admission, which may be medical or injury related. In France, patients admitted to the ED have an average age of 29, with a slight over-representation of men [31]. Injury patients tend to be younger and are more likely to be men. At the time of writing, two studies have explored reported feelings of pain or stress in the ED according to the criteria of age, gender, and diagnosis. A Canadian study has shown that, for patients experiencing pain when admitted to the ED for a medical issue, the intensity of the pain reported decreases with age, and that women reported higher levels of pain for three out of the six diagnoses selected by the authors of the study [32]. Another American study focused on reported pain and stress levels among patients with a pain level greater than 3 out of 10 [33]. In this second study, men and women report the same pain intensity, with the only difference being in stress level, which was reported as higher among women. As the patients in these two studies were selected based on their pain level, the study is not able to provide a global picture of the experiences of the ED patient population. Furthermore, to our knowledge, there is currently no other study that has quantified the emotions experienced in the ED.

The aim of this study is to better understand feelings of pain, stress, and emotion among the patients included in a vast multicentre study conducted within French EDs, according to their sociodemographic characteristics and reason for admission.

## 2. Materials and Methods

### 2.1. Population and Study Design

The study population was drawn from a randomised trial that measured the impact of a psychological intervention on the occurrence of post-traumatic stress disorder: SOFTER IV. In short, seven EDs from hospitals around France participated in the study. The study took place over 2 weeks in each centre. One week involved the intervention of a team of psychologists in addition to the usual care provided (intervention week), and during the other week, care was provided as usual (control week). Randomisation was based on the order of intervention and control weeks. The same team of psychologists visited all seven centres successively. The enrolment period for the study commenced on 18 October 2021 and lasted 9 months. Patients had to be over 18 years of age, fully conscious, and able to give their consent and agree to a telephone consultation 4 months after their discharge from ED. SOFTER’s inclusion criteria were compatible with the feelings study, and every eligible patient was recruited consecutively following ED admission. Patients were asked to fill out a questionnaire upon admission to the ED, after the patient was seen by the triage nurse, and before medical care was provided. This study included patients whose reason for ED admission was known, resulting in a population of 2846 patients out of a total of 2965 (Figure 1).

### 2.2. Feelings Recorded in the ED

The variables specific to the study on feelings were integrated into the SOFTER questionnaire. Patients completed the questionnaire containing all the study variables on a tablet, in most cases by themselves. When necessary or when requested by the patients, they were given the option of a paper version of the questionnaire. The interviewer could also complete the questionnaire for patients who were physically unable to do so. In this case, the interviewers simply read out the instructions provided on the self-assessment questionnaire.

Acute pain at entry to the ED was evaluated on a numerical rating scale (NRS) of 0 to 10. The Initiative on Methods, Measurement, and Pain Assessment in Clinical Trials (IMMPACT) endorses self-assessment as the gold standard for pain measurement [34]. In the case of people capable of self-assessment, the most commonly used tool is the NRS [35]. Widely used in the medical field, this scale may have originally been developed by Dr. Melzack and Dr. Torgerson in 1975 at McGill University in the first McGill pain questionnaire, with a version of the intensity scale ranging from 0 to 20 [36]. For stress, a scale based on the NRS was recently developed in a paediatric clinical context by Karvounides et al. [37]. The aim was to provide an easy-to-use assessment scale with a short administration time to assess current stress. This scale of 0 to 10 was used.

These variables were recorded continuously and grouped into four categories (no (0), low (1–3), moderate (4–5), and severe (6–10)). These categories are based on the principle of providing “priority” relief from the threshold of 4/10 mentioned in French emergency medicine recommendations for pain [7].

Emotions were surveyed using a methodology similar to that used for pain and stress, encouraging the participant to think about the presence of certain emotions and their intensity. This is made possible by the Genova Emotion wheel, created by Scherer [20]. As for pain and stress, it allows emotions to be measured in the moment in a short format. It is mainly used in experimental studies requiring comparison between experimental groups. Patients were specifically questioned on eight emotions (presence and intensity on a scale of 1 to 5) according to the Geneva Emotion Wheel, with an equal representation of positive and negative emotions [38]. Anger, fear, regret, and sadness were selected for the negative emotions, and relief, satisfaction, joy, and interest were selected for the positive emotions. These variables were recorded continuously and split into two categories (no or low (0–2) and strong (3–5)), allowing us to remain consistent with the 6–10 out of 10 classes used for pain and stress. Patients were asked if they felt any emotions other than those listed. These were categorised as either positive or negative, but their intensity was not evaluated.

### 2.3. Age, Gender, and Reason for ED Admission

Three patient characteristics were selected to describe the ED population and their feelings: age, as a continuous range, then grouped into two classes corresponding to the 3rd quartile of the distribution (<60 years, which accounts for 75% of the study population, ≥60 years), gender, and their reason for admission to the ED. The reason for admission was recorded into 15 predefined levels, supplemented by an open-ended question. These reasons were further grouped into two main categories: illness and injury. The illness category included all illnesses (heart issues, respiratory issues, neurological disorders, abdominal issues, digestive issues, headaches), and the injury category included all injury types (road accidents, falls, domestic accidents, sport accidents, work-related accidents, school accidents, attempted suicides, violence, leisure-related accidents). The answers provided in response to the open-ended question reserved for other possible reasons were allocated to one of these two categories.

### 2.4. Other Factors

Other factors described were the patient’s level of education (in number of years of study), divided into two categories (<12 years, ≥12 years) since in France the general secondary school diploma (*Baccalauréat*) is awarded after 12 years of schooling, employment (yes/no), self-perceived state of health on a scale of 1 to 5 and split into two categories (poor or bad vs. excellent, very good and good) and the existence, prior to this visit to the ED, of chronic pain, a persistent or recurring pain lasting more than 3 months.

### 2.5. Statistical Analysis

The sample size was calculated to demonstrate a reduced risk of post-traumatic stress disorder when psychological care is provided in the ED. Given the descriptive nature of our study, we considered that the sample size would enable us to describe feelings in detail according to age, gender, and reason for admission.

Patient characteristics and feelings were described generally and according to age, gender, and their reason for admission (injury or medical). The intensity of these feelings in relation to these various factors were compared using the Student’s *t*-test after testing the normality of the distributions. The means and standard deviations of negative emotions were broken down by specific reasons for seeking emergency care. We then calculated the proportion of patients reporting at least one strong negative emotion and the proportion reporting at least one strong positive emotion. To account for cases in which a patient may report several emotions, we examined the intensity of emotional valency, i.e., the negative or positive nature of emotions. The intensity of the highest negative emotion represented the intensity of the negative emotional valency, and the intensity of the highest positive emotion represented the intensity of the positive emotional valency. Correlations between experiences of pain, stress, and negative emotional valency were verified by calculating Pearson’s correlation coefficients. Finally, the adjusted associations between age, gender, and admission reason with pain, stress, negative emotional valency, and positive emotional valency were estimated by beta coefficients using linear regression models. Analyses were carried out using SAS V9.

## 3. Results

### 3.1. Characteristics of Patients Admitted to the ED

The population of 2846 adults admitted to the ED had an average age of 44 (median = 40), was made up of roughly equal numbers of men and women, and illness was the reason for their admission, reported by 62.6% of patients (Table 1).

The most common illnesses, in descending order, were abdominal (21.8%), neurological (17.3%), heart (9.0%), and respiratory (8.2%). Injury was the reason for admission for 37.4% of patients, specifically falls (21.1%), road accidents (18.6%), work-related accidents (17.2%), sports accidents (14.4%), and domestic accidents (14.2%) in descending order of frequency. Age, gender, and reason for admission characteristics of the patients recruited in each center are described in Appendix A.

Compared to patients aged 60 and over, patients aged 18–59 were significantly more likely to have been admitted to the ED for an injury-related reason, to have completed at least 12 years of study, to be in employment, to describe their own state of health as good and to not have any history of chronic pain (Table 1). Women were significantly more likely than men to have been admitted to the ED for illness, to not be in employment, to describe their own state of health as poor, and to have a history of chronic pain. Finally, in comparison with patients admitted for injury, patients admitted for illness were more likely to be women, had an average age of almost 6 years older, and were significantly more likely to not be in employment, to describe their own state of health as poor, and to have a history of chronic pain.

### 3.2. Patient Feelings at ED Admission

For all patients, the average pain intensity was 4.5 out of 10, with a median of 5 (Table 2).

Just over a third of patients (36.4%) reported a level of pain of strictly less than 4, and 42.3% reported severe pain (6–10). The average stress intensity was 3.4 out of 10, with a median of 3. More than half of patients (52.8%) reported a stress level of strictly less than 4, and 28.3% reported severe stress (6–10). The two most frequently reported high-intensity emotions (3–5) were fear (25.1%) and sadness (24.6%) (Table 2).

These were followed by regret (17.2%) and anger (14.5%). Slightly less than half the patients reported feeling at least one strong negative emotion (45.7%). The other negative emotions most commonly mentioned in response to the open-ended question were anxiety (2.3%), worry (1.8%), and lassitude (1.7%). As for high-intensity positive emotions, interest and relief were the most highly represented (27.9% and 27.8%, respectively), followed by satisfaction and joy (17.2% and 8.3%, respectively). Also, slightly less than half the patients reported feeling at least one strong positive emotion (46.9%). The other most commonly mentioned positive emotions were calm (0.4%) and tranquility (0.3%).

In all ED patients, feelings of pain, stress, and negative emotions were significantly correlated in pairs. The strongest correlations were observed between stress and the presence of at least one strong negative emotion (rho = 0.46, *p* < 0.0001), followed by pain and stress (rho = 0.23, *p* < 0.0001), and finally between pain and the presence of a strong negative emotion (rho = 0.17, *p* < 0.0001).

### 3.3. Feelings in Relation to Sociodemographic Factors

#### 3.3.1. Feelings in Relation to Patient Age

In comparison with patients aged 60 and over, patients between the ages of 18–59 reported a significantly higher intensity of pain and stress. They were also more likely to report fear and regret and less likely to report relief, satisfaction, and interest (Table 2).

#### 3.3.2. Feelings in Relation to Patient Gender

Women reported a significantly higher intensity of pain and stress than men. They were also more likely to report anger, fear, and sadness and less likely to report relief, joy, and interest (Table 2).

#### 3.3.3. Feelings in Relation to Reason for Admission

In comparison with patients admitted for illness, patients admitted for injury reported significantly higher intensities of pain. They were also more likely to report anger and regret (Table 2). The highest level of pain intensity was observed among road accident victims (m = 5.4; SD = 2.3) (Table 3).

Victims of violence were the most likely to report strong anger (51.2%), keeping in mind that these cases only represented 1.5% of injury admissions. Violence victims were also found to experience regret more often (32.6%). Patients admitted for illness reported significantly higher intensities of stress. They were more likely to report fear but also relief and interest. The highest stress levels were observed among patients admitted for a heart issue (m = 4.1; SD = 2.9). Patients admitted for a neurological issue, and patients admitted for violence were more likely to report fear (33.8% and 34.9%, respectively). In terms of positive emotions, relief was more likely to be reported among patients admitted for a digestive issue (37.1%), satisfaction for patients admitted for a digestive issue or for ear, nose, and throat issues (23.4% and 23.3%, respectively), and interest for patients admitted for a neurological issue (35.7%).

#### 3.3.4. Feelings in Relation to Age, Gender, and Reason for Admission

When taking into account the three characteristics of age, gender, and reason for ED admission in the multivariate regression model (Table 4), patients aged 18–59 reported significantly higher intensities of pain and stress than those aged 60 and over, but also a significantly lower intensity of positive emotional valency; women reported significantly higher intensities of pain, stress, and negative emotional valency than men, but also a lower intensity of positive emotional valency; patients admitted for injury reported higher intensities of pain and negative emotional valency than those admitted for illness and significantly lower intensities of stress and positive emotional valency.

## 4. Discussion

From the study population of 2846 adults admitted to the ED, all reasons combined, pain intensity upon entering the ED was measured at 4.5 out of 10 on average and 3.4 out of 10 for stress. Forty-six percent of patients described feeling at least one strong negative emotion, the most common being fear and sadness. Forty-seven percent of patients described feeling at least one strong positive emotion, the most common being interest and relief.

Pain was significantly higher among female patients under 60 admitted for injury than among male patients aged 60 or over admitted for illness. Stress was significantly higher among female patients under 60 admitted for illness than among male patients aged 60 or over admitted for injury. Emotions of negative valency were significantly higher among women admitted for injury than for men admitted for illness. Emotions of positive valency were significantly higher among men over 60 admitted for illness than for women under 60 admitted for injury.

There were positive but weak correlations between feelings, suggesting expressions of different states that were, therefore, worth studying separately.

### 4.1. Pain at ED Admission

In our study, the average level of pain reported by all patients admitted to the ED was 4.5 out of 10, far lower than the average level of 7.1 observed in the American study by Patel et al. [33]. The aim of our study was to describe various feelings, pain, stress, and emotional states, not excluding patients suffering from no or mild pain, which is often the case in the literature. Although Patel et al.’s study excludes 1035 patients who reported a pain level of less than or equal to 3, the findings are fairly consistent with the results of our study (mean = 6.5, SD = 1.7, n = 1811 patients). They are comparable with the results of other studies excluding non-pain sufferers. Thornton et al. observe a pain intensity median of 6, a median which is also found in our own study, after excluding 474 patients not suffering from pain [39]. In their study, Guéant et al. observed an average pain level at ED admission of 5.2 (SD = 2.5) [3]; in our study, after excluding 474 non-pain sufferers, we found an average of 5.5 (SD = 2.3).

Pain was significantly higher among female patients, patients under 60, and patients admitted for injury.

Just like in Patel et al., women report experiencing a higher level of pain than men, even after excluding patients with a pain level of 0–3. The different biological and psychological mechanisms of men and women may explain this higher perception of pain among women [40]. According to Social Learning Theory, behaviour is not simply a product of personal and environmental influences; these components interact as mutual determinants of each other. Men and women have different levels of self-efficacy, which could lead to different reactions to the sensation of pain [41]. In our study, patients under 60 years old reported a higher level of pain than over-60s. This same age effect was observed in Daoust et al., particularly for diagnoses not necessarily linked to age (renal colic, pancreatitis, appendicitis, and headache/migraine) [32]. These authors suggest that this lower perception of pain in older patients could be explained by the inflammatory processes and reduction in pain receptor activity associated with older age [42]. Finally, pain intensity was higher among patients admitted for injury rather than for illness, particularly among road accidents and fall victims. If we exclude non-pain sufferers, as seen in Thornton et al., we also do not find any significant difference in pain between the two reasons for admission, injury, or illness [39]. In other words, the percentage of non-pain sufferers among patients admitted to the ED for illness is higher than that observed among those admitted for injury (22% vs. 8%). When it comes to the type of accident leading to injury, two studies are consistent with our results, with a pain level of 6.3 out of 10 after a fall [43] and 6.4 after a road accident [44].

### 4.2. Stress at ED Admission

The average stress level observed in our study was 3.4 out of 10, 2 points lower than the level measured by Patel et al. (5.9). In contrast to pain, even if we exclude patients reporting pain of less than or equal to 3, we find a lower average stress level (3.8, SD = 3.1), even though the age and gender characteristics of the two study populations are fairly similar.

Stress was higher among female patients, patients under 60, and patients admitted for illness.

Similarly to Patel et al., we found a higher self-reported stress level in women than in men, even after excluding patients describing their pain level as 0–3 to allow for comparability with the Patel population). This difference is confirmed by Matud et al. in their study involving more than 2800 individuals [45]. In this study, women reported a higher level of stress than men despite having experienced a similar number of life events over the past several years. They had a more negative view of these life events and felt less in control of these situations. They were also found to be more sensitive to events associated with family and health, while the main causes of worry for men involved relationships, financial problems, or work. Women are also believed to adopt coping mechanisms based on emotions and avoidance rather than rational adaptation and detachment and often find emotional release more easily than men.

Regarding the effect of age on stress levels, one study on daily stress lists age as a resilience factor for stress associated with perceived control [46]. Trouillet et al. highlighted a change in approach when it comes to coping mechanisms among older people rooted in self-efficacy and social support fulfillment [47].

Finally, stress levels were higher among patients admitted for illness than for injury, particularly among those suffering from heart problems or digestive issues. It is hard to say whether these patients suffered from chronic stress, but the link between stress and cardiovascular disease is well known [48,49]. As for digestive issues, ulcers are also associated with chronic stress [19,50]. Patients suffering from these pathologies may be more likely to have a stress reaction to a negative stimulus.

### 4.3. Emotions at ED Admission

It is difficult to compare our results on the intensities of negative and positive emotions, or even the mere presence of such emotions, since, to our knowledge, this study is the first to quantify a broad spectrum of emotions in a clinical context. Patel et al.’s study focused on anxiety, with an average intensity of 5.8 out of 10 among patients with a pain level of 4 out of 10 or over. Amstadter & Vernon, on the other hand, propose a retrospective study of university students and their peri- and post-traumatic emotions triggered by an event involving physical or sexual assault, a road accident, injury, or illness [51]. Students reported, in descending order of intensity, emotions of fear, sadness, anger, guilt, and shame for the peri-traumatic phase we are examining in this present study. In our study, fear and sadness were also seen to be the highest-intensity emotions. A study of patients admitted to EDs for self-harm also highlighted the high level of anger among these patients [52]. It is also interesting to note the existence of strong positive emotions on entering the ED, which could correspond, for some patients, to an initial sense of relief with regard to the reason that brought them to the ED. The exploration of the co-existence of positive and negative emotional valences may help to identify specific patient profiles.

Negative emotions were higher among female patients and patients admitted for injury. Positive emotions were higher among male patients, patients over 60, and patients admitted for illness.

Our results suggest a higher level of negative emotions in women than in men and a higher level of positive emotions in men. Patel et al.’s study identifies a higher level of self-reported anxiety among women [33]. These results are also consistent with the study by Matud et al., which reveals coping mechanisms among women involving a higher expression of negative emotions and lower concern for health issues among men [45].

Our study only observed age as a contributing factor to the intensity of positive valency emotions. These results are consistent with a study conducted during the SARS pandemic in which people aged over 55 were found to be more likely to adopt a coping strategy based on adaptation or emotion, which may consist of reappraising positively aversive situations, as opposed to problem-focused coping mechanisms, which aim to modify the situation, more commonly observed in young people [53].

Patients admitted for injury reported negative emotions of a higher intensity than those admitted for illness, while those admitted for illness reported positive emotions of a higher intensity than those admitted for injury. Nevertheless, the specific nature of these emotions and reasons for admission put these results in perspective. Anger and regret were particularly pronounced among injury patients who had suffered violence or road accidents. Meanwhile, fear was found to be particularly high among patients suffering from a neurological issue. In the review conducted by Anestis et al., various studies have pointed toward a greater feeling of distress when being diagnosed with a neurological condition [54]. Strong negative feelings were also reported for patients admitted for psychiatry-related reasons: suicide attempts and mental health issues such as anxiety or depression. This highlights a need for medical care that goes beyond the somatic for patients of this type.

### 4.4. Strengths and Weaknesses

The population used for this study was drawn from seven hospital EDs, with various socioeconomic factors at play for each hospital context. Only excluding life-threatening emergencies and patients who did not speak French, our inclusion criteria allowed us to recruit a very high number of patients, representing a significant percentage of emergency patients.

The originality of this study resides in its description of emotions experienced in the ED. Four negative emotions and four positive emotions were chosen to compose an emotion wheel inspired by that proposed by Scherer. The inclusion of an open-ended question allowed us to collect data on other emotions. The range of emotions to choose from allowed patients to describe their emotional state, and very few reported another emotion. It also highlights the value of asking patients about their positive emotions, since in our study, more than half of patients reported at least one positive emotion, most frequently relief (39%). Their presence in an ED context may be a sign that previous negative emotions are giving way to positive ones. However, the dynamics of emotions could not be observed in this study. Nonetheless, their presence and, in particular, feelings of relief and interest could be interesting clinical indicators. Interest could presage success in the therapeutic alliance with caregivers and better adherence to the care pathway. As for appeasement, its presence is reassuring for both patients and health professionals. It may be the first step towards the resolution of the health problem that brought the patient to the service. Evaluating the association between the presence of positive emotions and recovery would provide further insight into their clinical relevance to health outcomes following a visit to the ED. Our study shows the feasibility and interest of assessing emotional state, in particular with an emotion wheel, at the entrance to the emergency department. This opens the way for future research testing the relationship between changes in emotional state in the ED and patient-perceived quality of care, in line with the patient-reported outcome development.

The Geneva Emotion Wheel is used internationally, particularly in experimental studies, because it imposes emotion forced-choice responses, and so ensures comparability between experimental groups. Its use in clinical studies is relatively rare. However, it offered the same advantages for experimental studies, i.e., control of comparability between patients, which is necessary for future studies on health consequences. As for pain and stress, it also enabled precise quantification of the emotions assessed as part of this study. Moreover, the Geneva Emotion Wheel is a flexible tool that can be adapted to the needs of each study, as the emotional items can be selected in part. To select items in other languages or cultures, it may be necessary to use the Geneva Affect Label Coder (GALC), available in English, German, and French [55]. The Geneva Emotion Wheel requires symmetry within the different dimensions and, thus, for valency, the presence of the same number of positive and negative emotions. Apart from the primary emotions of fear, sadness, and anger for the negative emotions and joy for the positive emotions, which are initially present on the wheel, the emotions of regret, relief, satisfaction, and interest were specifically included in the study’s emotion wheel, as they were assumed to correspond to emotions that could be aroused by a health event for “regret”, or a care situation for these other positive emotions. It is, therefore, possible that they are not adapted to other contexts, such as other care situations or countries where common emotional expression differs from France.

This study is based on self-reported feelings. It is, therefore, possible that some patients did not report their feelings accurately, for example, due to a lack of emotional awareness. As this was the first quantitative study of emotions in the ED, the emotion wheel of this study was based on a study on anger, which proposed a 5-point scale [52]. It is possible, however, that the change from a 10-point scale to a 5-point scale may have confused patients. It would therefore have been preferable to homogenise the scales by adopting a 10-point scale for all feelings. Furthermore, individual differences in experiencing emotions and relationships (e.g., attachment style, primary emotional traits, emotion regulation, alexithymia, anxiety, and depression) were not measured, as this would have shed further light on the declarations of feelings.

The three feeling categories of pain, stress, and emotion vary depending on age, gender, and reason for admission. They were also relatively uncorrelated, the strongest correlation being 0.46 between self-reported stress and the most intense negative emotion. Therefore, a patient’s emotional state cannot be fully determined by their stress level.

The data available for this study did not include the diagnosis provided at discharge from the ED. Although being able to describe the feelings in relation to this diagnosis would have been interesting, the fact that they are described in relation to the reason for admission, information that is immediately available, makes our results particularly useful in terms of the potential impact of these feelings after admission to an ED.

It would also have been interesting to have access to information on local contexts, such as the atmosphere in the ED at the time of admission waiting times and the specific characteristics of each department.

Feelings may have been influenced by the social support patients received during their time in the ED. However, this information was also not covered as part of our data collection.

## 5. Conclusions

The present study highlights the strong presence of pain, stress, and emotions experienced by patients admitted to the ED. These feelings vary according to sociodemographic and health characteristics, and they are correlated, albeit weakly. The management of feelings in the ED is essentially focused on pain, which is, of course, essential. However, our results suggest the need to also manage stress and emotions and encourage further work to define what types of management of these feelings could complement current protocols. It could also be useful to monitor these feelings over time and assess how they are impacted by the context in which they are expressed.

## Figures and Tables

**Figure 1 healthcare-13-00500-f001:**
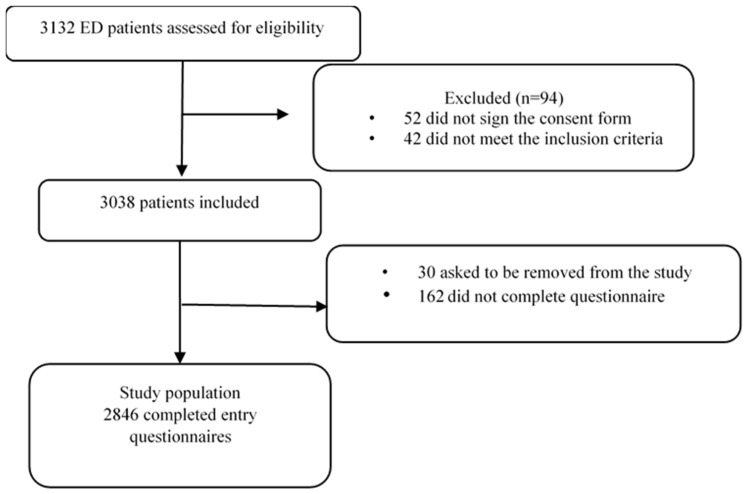
Study flowchart.

**Table 1 healthcare-13-00500-t001:** Patient characteristics at ED admission.

	All Patients	Age			Gender			Reason for Admission		
	n = 2846	18–59 n = 2140	≥ 60 n = 706	*p*-Value ^(1)^	Men n = 1445	Women n = 1401	*p*-Value ^(1)^	Illness n = 1782	Injury n = 1064	*p*-Value ^(1)^
Age, % 18–59	75.2	-	-	-	75.0	75.4	0.8	72.3	80.1	<0.0001
Gender, % male	50.8	50.6	51.1	0.8	-	-	-	46.6	57.8	<0.0001
Reason for admission, % injury	37.4	39.8	30.0	<0.0001	42.6	32.0	<0.0001	-	-	-
≥12 years of education, %	65.9	71.8	48.0	<0.0001	64.6	67.3	0.1	64.7	67.9	0.08
In employment, %	66.1	83.8	12.5	<0.0001	68.1	64.0	0.02	62.9	71.4	<0.0001
Poor perceived health, % ^(2)^	21.8	17.7	34.3	<0.0001	18.1	25.6	<0.0001	27.4	12.3	<0.0001
History of chronic pain, % ^(3)^	32.5	30.5	38.4	0.0001	27.7	37.3	<0.0001	38.0	23.1	<0.0001

^(1)^ Chi-2; ^(2)^ poor or bad vs. excellent, very good, and good; ^(3)^ prevalent chronic pain sufferers who answered “yes” to the question: “Do you have pain that has been persistent or recurring for more than 3 months?”.

**Table 2 healthcare-13-00500-t002:** Pain, stress, and emotions at ED admission according to age, gender, and reason for admission, n = 2846.

	Pain			Stress			Negative Emotions				Positive Emotions			
							Anger	Fear	Regret	Sadness	Relief	Satisfaction	Joy	Interest
	m (SD)	Moderate [4–5] %	Severe [6–10] %	m (SD)	Moderate [4–5] %	Severe [6–10] %	Strong [3–5] %							
All patients														
	4.5 (3.0)	21.3	42.3	3.4 (3.1)	18.9	28.3	14.5	25.1	17.2	24.6	27.8	17.2	8.3	27.9
Age														
18–59	4.8 (2.8)	22.1	45.1	3.5 (3.1)	18.7	28.9	14.7	26.5	18.0	23.9	25.0	16.2	8.2	26.6
60+	3.8 (3.3)	18.8	33.9	3.2 (3.1)	19.4	26.5	13.9	20.8	14.7	26.6	36.3	20.5	8.5	31.6
*p*-value *	<0.0001			0.04			0.6	0.003	0.04	0.1	<0.0001	0.008	0.8	0.01
Gender														
Men	4.2 (2.9)	21.4	37.2	2.8 (2.9)	17.2	21.1	13.2	17.3	16.8	18.1	29.5	18.5	10.4	29.5
Women	4.9 (3.0)	21.3	47.5	4.0 (3.1)	20.6	35.7	15.8	33.1	17.6	31.3	26.0	16.0	6.1	26.1
*p*-value *	<0.0001			<0.0001			0.04	<0.0001	0.5	<0.0001	0.03	0.08	<0.0001	0.04
Reason for admission														
Illness	4.4 (3.1)	20.5	40.6	3.7 (3.1)	20.4	31.1	11.7	28.4	12.4	25.2	29.7	18.2	8.3	31.1
Injury	4.9 (2.7)	22.7	45.2	3.0 (3.0)	16.3	23.6	19.3	19.5	25.3	23.5	24.5	15.7	8.2	22.5
*p*-value *	<0.0001			<0.0001			<0.0001	<0.0001	<0.0001	0.3	0.003	0.09	0.9	<0.0001

m = mean, (SD) = standard deviation; * *t*-test mean comparisons (pain and stress) and Chi-2 test for proportion comparisons (emotions).

**Table 3 healthcare-13-00500-t003:** Intensity of feelings of pain, stress, and negative emotions by detailed reason for admission, n = 2846.

	n	Pain	Stress			Negative Emotions				Positive Emotions			
						Anger	Fear	Regret	Sadness	Relief	Satisfaction	Joy	Interest
		m (SD)	Severe [6–10] %	m (SD)	Severe [6–10] %	Strong [3–5] %							
Reason for illness													
Abdominal	389	5.5 (2.7)	54.2	3.6 (3.0)	30.8	8.2	29.1	8.5	23.1	28.5	17.2	8.0	28.5
Neurological	308	2.6 (3.0)	20.4	3.7 (3.1)	31.2	12.3	33.8	14.0	30.2	30.8	19.2	6.8	35.7
Cardiological	161	3.4 (2.7)	25.4	4.1 (3.0)	34.2	14.3	28.6	9.9	23.0	32.9	19.9	7.5	31.1
Respiratory	147	4.0 (2.9)	35.4	3.7 (3.0)	31.3	8.2	23.8	10.2	25.8	28.6	21.1	12.2	34.0
General medicine	130	3.7 (3.3)	34.6	3.1 (3.3)	27.7	10.0	30.8	13.1	24.6	26.1	15.4	9.2	28.5
Digestive	124	4.9 (2.9)	47.6	3.8 (3.0)	33.1	15.3	34.7	12.9	22.6	37.1	23.4	13.7	29.8
Headache	113	5.7 (2.8)	61.1	4.1 (3.0)	37.2	12.4	20.3	17.7	28.3	23.9	15.9	4.4	28.3
Rheumatological	84	6.9 (2.1)	76.2	3.8 (3.3)	38.1	14.3	27.4	15.5	25.0	22.6	10.7	7.1	20.2
Infection	76	5.2 (2.8)	51.3	3.3 (3.0)	25.0	10.5	25.0	17.1	25.0	34.2	19.7	11.8	26.3
Ear, nose, throat	43	3.4 (3.0)	25.6	3.0 (3.1)	20.9	11.6	18.6	n = 0	11.6	30.2	23.3	9.3	30.2
Pain management	21	5.9 (2.3)	n = 15	4.1 (3.5)	n = 7	n = 4	n = 4	n = 2	n = 7	n = 10	n = 7	n = 2	n = 9
Dermatological	20	3.8 (2.3)	n = 4	2.9 (3.0)	n = 5	n = 3	n = 4	n = 3	n = 3	n = 6	n = 6	n = 3	n = 10
Psychological	12	2.2 (2.8)	n = 1	5.1 (3.8)	n = 7	n = 4	n = 6	n = 5	n = 8	n = 6	n = 2	n = 0	n = 7
Other	154	3.9 (3.1)	31.8	3.3 (3.1)	25.3	13.6	24.7	16.2	23.4	27.3	12.3	5.2	33.1
Reason for injury													
Fall	224	5.2 (2.8)	49.5	3.6 (3.1)	32.1	17.9	24.5	25.4	30.8	26.3	19.2	8.5	21.9
Road accident	198	5.4 (2.3)	55.0	3.2 (3.0)	25.8	22.2	28.3	30.3	31.3	31.3	14.1	6.1	20.7
Work-related	183	4.8 (2.7)	43.7	2.3 (2.8)	15.3	17.5	13.7	16.9	12.0	17.5	18.6	11.5	20.8
Domestic	151	4.3 (3.0)	37.7	2.9 (3.1)	23.2	15.9	11.9	22.5	20.5	28.5	15.2	7.3	22.5
Sports	153	4.6 (2.5)	39.2	2.2 (2.5)	11.1	14.4	9.1	22.9	8.5	19.6	11.1	11.8	24.2
Leisure-related	55	5.1 (2.5)	56.4	2.4 (2.5)	14.5	7.3	12.7	30.9	18.2	25.4	12.7	5.4	34.5
Suicide attempt	19	3.9 (3.5)	n = 7	6.9 (3.0)	n = 14	n = 1	n = 11	n = 15	n = 16	n = 2	n = 2	n = 1	n = 4
Violence	43	4.7 (2.7)	37.2	4.9 (3.5)	46.5	51.2	34.9	32.6	53.5	23.3	16.3	2.3	23.3
Other *	38	4.1 (2.4)	26.3	2.5 (2.8)	15.8	13.2	18.4	15.8	10.5	23.7	15.8	2.6	18.4

* School accidents were added to the other category for injury patients; m (SD) = mean and standard deviation; N.B.: percentages of severe or strong feelings in categories with fewer than 25 patients were not reported.

**Table 4 healthcare-13-00500-t004:** Patient characteristics in relation to feelings at ED admission, multivariate linear regression, n = 2846.

Patient Characteristics	Pain		Stress		NE *		PE *	
	Beta Coeff. (CI)	*p*-Value	Beta Coeff. (CI)	*p*-Value	Beta Coeff. (CI)	*p*-Value	Beta Coeff. (CI)	*p*-Value
≥60 vs. <60 years	0.97 (0.73; 1.20)	<0.0001	0.33 (0.01; 0.58)	0.01	0.01 (−0.15; 0.17)	0.9	−0.46 (−0.30; −0.63)	<0.0001
Women vs. men	0.75 (0.54; 0.97)	<0.0001	1.19 (0.96; 1.41)	<0.0001	0.66 (0.52; 0.80)	<0.0001	-0.15 (−0.01; −0.29)	0.03
Injury vs. illness	0.54 (0.31; 0.76)	<0.0001	−0.54 (−0.31; −0.78)	<0.0001	0.20 (0.10; 0.34)	0.007	-0.36 (−0.22; −0.51)	<0.0001

* NE = negative emotion; PE = positive emotion; CI = confidence interval.

## Data Availability

Data is available on reasonable request.

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
