# Peer review of "Feelings of Patients Admitted to the Emergency Department"

_healthcare, 2025, doi:10.3390/healthcare13050500_

Round 1

Reviewer 1 Report

Comments and Suggestions for Authors
  1. Introduction:

    The introduction offers sufficient background information but would benefit from the inclusion of additional global studies to enhance its contextual breadth. Incorporating findings from countries beyond France could provide valuable comparative insights.

    The novelty and significance of the study should be articulated more prominently and earlier in the introduction.

    Research Design:

    The research design appropriately addresses the study's questions. However, the manuscript should provide a detailed explanation of the sample size determination and its adequacy.

    Clarify how the randomization process within the SOFTER IV study ensured a representative sample across various hospital settings.

    Methods:

    The methodology is adequately described. However, a brief justification for selecting specific data collection tools, such as the Geneva Emotion Wheel, should be included.

    Elaborate on the rationale for grouping emotional intensities and categories.

    Results:

    The results are presented clearly, but all figures and tables should be made fully self-explanatory with comprehensive captions.

    Consider providing a more detailed breakdown of demographic data, focusing on regional or socioeconomic variations, if such data is available.

    Conclusions:

    The conclusions are consistent with the results. However, a more in-depth discussion of their potential implications for international emergency department policies and practices is recommended.

    English Language:

    The manuscript is generally well-written but would benefit from professional language editing to improve clarity, flow, and eliminate minor grammatical errors.

    Additional Recommendations:

    • Include more detailed statistical analyses in supplementary materials to enhance transparency.

    • Discuss limitations regarding the cultural specificity of the Geneva Emotion Wheel and its relevance to populations outside France.

    • Explicitly highlight any unexpected findings and their broader implications in the discussion section.

Comments on the Quality of English Language

The manuscript is generally well-written but would benefit from professional language editing to improve clarity, flow, and eliminate minor grammatical errors.

Author Response

  1. Introduction:

Comment 1: The introduction offers sufficient background information but would benefit from the inclusion of additional global studies to enhance its contextual breadth. Incorporating findings from countries beyond France could provide valuable comparative insights.

Response 1: We have carried out an international review and perhaps quoted a few more French figures to compare our participants within the same context. The vast majority of our references are foreign to France. However, at the request of reviewer 2, we have added a review of the literature on patient experience in emergency departments. Our results are then discussed in relation to the studies of Patel et al, Thornton et al, and Guéant et al. cited in the introduction. 

Comment 2: The novelty and significance of the study should be articulated more prominently and earlier in the introduction.

Response 2: Earlier, in the introduction, we added a sentence that announces the originality of our study on feelings in the emergency department:

Lines 48-50:

“These feelings experienced in the ED, particularly stress and emotions, deserve special attention, just like with pain, which is more often explored.”

Research Design:

Comment 3: The research design appropriately addresses the study's questions. However, the manuscript should provide a detailed explanation of the sample size determination and its adequacy.

Response 3: The sample size was calculated to demonstrate a reduced risk of PTSD when psychological care is provided in the ED. Given the descriptive nature of our study, we considered that the sample size would enable us to describe feelings in detail according to age, gender and reason for admission.

We added this comment on the sample size and statistical part of the methodology, section 2.5, lines 198-201.

Comment 4: Clarify how the randomization process within the SOFTER IV study ensured a representative sample across various hospital settings.

Response 4: The randomization process was not based on study participants but on intervention weeks within the study. The randomized trial took place over two weeks in each center: an intervention period with the presence of psychologists in addition to the usual team, and a control period with only the usual emergency team, the general medicine team. The weeks were randomly ordered, and a wash-out period was performed in between. During the week of intervention, the psychologists first attended the patients identified by the care teams as being most in need of psychological intervention, and then attended to as many patients as possible.

The 7 centers were chosen to represent a wide diversity of emergency departments in France. During each of the 2 weeks, every eligible patient was approached.

We have now added the following statement to the article that respond to reviewer 1 and 2 comments on the methodology (lines 122-127):

“The study took place over 2 weeks in each centre. One week involved the intervention of a team of psychologists in addition to the usual care provided (intervention week), and during the other week care was provided as usual (control week). Randomisation was based on the order of intervention and control weeks. The same team of psychologists visited all 7 centres successively. The enrolment period for the study commenced on 18 October 2021 and lasted 9 months.”

And lines 129-130:

“SOFTER's inclusion criteria were compatible with the feelings study, and every eligible patient were recruited consecutively following ED admission.”

As additional data (also requested by reviewer 2), we provide a table describing the age, gender, and raison for admission characteristics of the patients recruited in each centre.

Appendix 1. Patient characteristics at ED admission for each centre

ED 1

ED 2

ED 3

ED 4

ED 5

ED 6

ED 7

Number of patients, n

991

253

564

487

195

166

190

Age, % 18-59

76.3

62.1

72.0

78.8

83.6

69.3

83.7

Gender, % male

53.4

47.4

50.2

53.8

44.6

48.8

43.7

Reason for admission, % injury

35.9

35.2

36.3

47.6

30.3

30.1

38.4

Note: ED 1=Bordeaux University Hospital ; ED 2=Libourne Hospital ; ED 3=Edouard Herriot Hospital, HCL ; ED 4=Toulouse University Hospital ; ED 5=Teaching Hospital Louis Mourrier, AP-HP ; ED 6=Beaujon Hospital ; ED 7=Lariboisière Hospital, AP-HP.

Methods:

Comment 5: The methodology is adequately described. However, a brief justification for selecting specific data collection tools, such as the Geneva Emotion Wheel, should be included.

Response 5: Emotions were surveyed using a methodology similar to that used for pain and stress, encouraging the participant to think about the presence of certain emotions and their intensity. Moreover, by asking about specific emotions, the Geneva Wheel enables comparability between participants for studies on health consequences, but also offers the possibility of mentioning an emotion not proposed as a free response, which meant we could check that we hadn't missed an important emotion to include in the wheel.

We have now added the following statement to the article (lines 164-166):

« Emotions were surveyed using a methodology similar to that used for pain and stress, encouraging the participant to think about the presence of certain emotions and their intensity”

Comment 6: Elaborate on the rationale for grouping emotional intensities and categories.

Response 6: We haven't found any firm recommendations for grouping emotion intensities. The considerations seem to be adapted to the studies' objectives. Given the particular context of the emergency room, we felt it was coherent to focus on emotions of higher intensity, and thus to homogenize treatments between pain, stress and emotion. The 3-5 out of 5 class for emotions is comparable to the 6-10 out of 10 class used for stress and pain.

We added this detail line 174-175:

“allowing us to remain consistent with the 6-10 out of 10 class used for pain and stress”

Results:

Comment 7: The results are presented clearly, but all figures and tables should be made fully self-explanatory with comprehensive captions.

Response 7: We've added captions to tables 1, 2, 3, 4, 5 and 6, and also made a few additional corrections to make them more readable.

Table 1: we have added the “Age, % 18-59” line, which was an oversight in the description.

Table 2: we have added the total number of participants

Table 6: in accordance with the text (lines 317-325), we have reversed the direction of relative risk, taking patients aged 60+ as the reference. The results section has been modified accordingly.

Comment 8: Consider providing a more detailed breakdown of demographic data, focusing on regional or socioeconomic variations, if such data is available.

Response 8: We propose an additional data table describing the age, gender, and reason for admission of patients for each centre.

Appendix 1. Patient characteristics at ED admission for each centre

ED 1

ED 2

ED 3

ED 4

ED 5

ED 6

ED 7

Number of patients, n

991

253

564

487

195

166

190

Age, % 18-59

76.3

62.1

72.0

78.8

83.6

69.3

83.7

Gender, % male

53.4

47.4

50.2

53.8

44.6

48.8

43.7

Reason for admission, % injury

35.9

35.2

36.3

47.6

30.3

30.1

38.4

Note: ED 1=Bordeaux University Hospital ; ED 2=Libourne Hospital ; ED 3=Edouard Herriot Hospital, HCL ; ED 4=Toulouse University Hospital ; ED 5=Teaching Hospital Louis Mourrier, AP-HP ; ED 6=Beaujon Hospital ; ED 7=Lariboisière Hospital, AP-HP.

Conclusions:

Comment 9: The conclusions are consistent with the results. However, a more in-depth discussion of their potential implications for international emergency department policies and practices is recommended.

Response 9: The conclusion has been extensively rewritten to meet your request, as well as the request of reviewer 2 to focus more on the results of this research.

                            Here is the new paragraph for the conclusion, lines 508-516:

“The present study highlights the strong presence of pain, stress and emotions experienced by patients admitted to the ED. These feelings vary according to socio-demographic and health characteristics, and they are correlated, albeit weakly. The management of feelings in the ED is essentially focused on pain, which is of course essential. However, our results suggest the need to also manage also stress and emotions and encourage further work to define what types of management of these feelings could complement current protocols. It could also be useful to monitor these feelings over time, and asses how they are impacted by the context in which they are expressed.”

We've added a discussion point on correlations between feelings, lines 345-346:

“There were positive but weak correlations between feelings, suggesting expressions of different states that were therefore worth studying separately”

English Language:

Comment 10: The manuscript is generally well-written but would benefit from professional language editing to improve clarity, flow, and eliminate minor grammatical errors.

Response 10: We're sorry the reviewer felt that our writing style didn't facilitate reading. This is certainly more a question of style of expression than English language, as we wrote our manuscript in French and it was translated by a professional native English translator. The translation certificate will be transmitted. We are sending our article back to the translator, with its modifications, to ensure that there are no grammatical errors and for her to make some style changes.

On her advice we have changed the title to “Feelings of patients admitted to the emergency department” and made some changes in the text.

Additional Recommendations:

    • Comment 11: Include more detailed statistical analyses in supplementary materials to enhance transparency.

Response 11: We propose an additional data table describing the age, gender and raison for admission of patients at each centre.

We propose also the description of positive feelings by detailed reason for admission in Appendix 2:

Appendix 2. Average intensity of positive emotions by detailed reason for admission

Intensity of positive emotions, mean (SD)

N

Appeasement

Contentment

Joy

Interest

Reason for illness

Abdominal

389

1.2 (1.7)

0.8 (1.5)

0.4 (1.1)

1.2 (1.7)

Neurological

308

1.3 (1.7)

0.8 (1.5)

0.3 (1.1)

1.4 (1.9)

Cardiological

161

1.4 (1.7)

0.9 (1.6)

0.3 (1.0)

1.4 (1.8)

Respiratory

147

1.3 (1.8)

0.9 (1.6)

0.5 (1.3)

1.4 (1.8)

General medicine

130

1.1 (1.6)

0.6 (1.4)

0.4 (1.2)

1.2 (1.7)

Digestive

124

1.5 (1.7)

0.9 (1.6)

1.6 (1.3)

1.2 (1.8)

Headache

113

1.1 (1.6)

0.6 (1.3)

0.2 (0.7)

1.1 (1.7)

Rheumatological

84

1.0 (1.5)

0.6 (1.2)

0.3 (1.0)

0.9 (1.7)

Infection

76

1.3 (1.6)

0.8 (1.5)

0.6 (1.3)

1.0 (1.7)

ENT

43

1.3 (1.8)

0.9 (1.6)

0.4 (1.2)

1.2 (1.9)

Pain management

21

1.9 (1.8)

1.5 (1.9)

0.3 (0.9)

1.8 (2.1)

Dermatological

20

1.1 (1.8)

1.1 (1.7)

0.5 (1.3)

2.1 (1.9)

Psychological

12

2.2 (2.2)

0.7 (1.2)

-

2.6 (2.4)

Other

154

1.3 (1.8)

0.6 (1.4)

0.3 (0.9)

1.4 (1.9)

Reason for injury

Fall

224

1.1 (1.6)

0.8 (1.5)

0.3 (1.1)

0.9 (1.5)

Road accident

198

1.2 (1.6)

0.5 (1.3)

0.2 (0.9)

0.9 (1.7)

Work-related

183

0.9 (1.5)

0.8 (1.5)

0.5 (1.4)

0.9 (1.6

Domestic

151

1.2 (1.6)

0.6 (1.3)

0.3 (0.9)

1.0 (1.7)

Sports

153

0.9 (1.4)

0.5 (1.2)

0.4 (1.1)

1.0 (1.6)

Leisure-related

55

1.1 (1.7)

0.6 (1.4)

0.2 (1.0)

1.5 (1.9)

Suicide attempt

19

0.8 (1.5)

0.4 (1.1)

0.3 (0.9)

1.1 (1.6)

Violence

43

1.0 (1.7)

0.7 (1.5)

0.07 (0.4)

1.0 (1.7)

Other*

38

1.0 (1.7)

0.8 (1.6)

0.2 (0.9)

0.8 (1.5)

*School accidents were added to the other category for injury patients

SD=standard deviation; ENT= Ear, nose, and throat

We have also clarified multivariate models in the statistics section (2.5), lines 214-217:

    • Comment 12: Discuss limitations regarding the cultural specificity of the Geneva Emotion Wheel and its relevance to populations outside France.

Response 12: We added a critical comment and justification of this tool to the discussion, lines 479-488:

“The Geneva Emotion Wheel is used internationally, particularly in experimental studies, because it imposes emotion forced-choice response and so ensures comparability between experimental groups. Its use in clinical studies is relatively rare. However, it offered the same advantages for experimental studies, i.e. control of comparability between patients, which is necessary for future studies on health consequences. As for pain and stress, it also enabled precise quantification of emotions assessed as part of this study. Moreover, the Geneva Wheel is a flexible tool that can be adapted to the needs of each study, as the emotional items can be selected in part. To select items in other languages or cultures, it may be necessary to use the Geneva Affect Label Coder (GALC), available in English, German, and French.”

    • Comment 13: Explicitly highlight any unexpected findings and their broader implications in the discussion section.

Response 13: The present study was a cross-sectional one, describing feelings in the emergency department as a function of socio-demographic and health characteristics. We therefore had no particular hypothesis as to the expected results. Nevertheless, we were surprised by the strong presence of positive emotions in emergency departments. Further work on emotional valences and the co-existence of positive and negative valences would enable us to identify specific patient profiles.

We added the following sentence in the discussion, lines 434-437:

“It is also interesting to note the existence of strong positive emotions on entering the ED, which could correspond, for some patients, to an initially sense of relief with regard to the reason that brought them to the ED. The exploration of the co-existence of positive and negative emotional valences may help to identify specific patient profiles.” 

Comments on the Quality of English Language

Comment 14: The manuscript is generally well-written but would benefit from professional language editing to improve clarity, flow, and eliminate minor grammatical errors.

Response 14: We have sent again the manuscript with the revisions to the translator to make sure there are no grammatical errors and improve clarity.

Reviewer 2 Report

Comments and Suggestions for Authors

Thank you for the invitation to review this interesting study exploring the emotions perceived by patients during ED attendance. I recommend some revisions, mostly minor, based on my comments below:

Line 15 'interviewed' - that word made me expect to see a qualitative study, whereas this was a survey / instrument administration with quantitative analysis. Suggest avoid 'interview'.

Line 62 typo, 'addiction'

Line 64-65 re no other study documenting emotions experienced by ED patients. I am aware of a growing literature base for ED patient experience (including emotions and perceptions) and its measurement:
https://doi.org/10.1016/j.ienj.2009.05.004
https://doi.org/10.1177/2374373517731359

(I advise the authors to conduct a literature review)

If you mean something by 'emotion' that is distinct from experience otherwise, suggest include an accessible definition in the Introduction.

Line 97-98 re there being no other studies exploring feelings experienced in the ED. Again, I am not at all convinced by this statement as I am aware of a fairly substantial body of qualitative evidence stemming from ED enquiry. 

Line 108 on. The enrolment period lasted 9 months, but recruitment was carried out over two weeks. Are you saying that the intervention / control alternated every week for 9 months, or did you recruit at each site for only two weeks during the 9 months? I was not clear but imagine the latter based on overall sample.

Line 110 please state explicitly whether you approached every eligible patient consecutively registering at the seven departments during this enrolment period, or whether the 3132 patients assessed for eligibility relates to some other sampling frame e.g. the opportunistic availability of researchers etc. This is important to know so that readers can make some judgement as to the comprehensiveness of sampling (ie, is it possible that people with certain characteristics were not approached or assessed for eligibility).

Line 121 how were feelings recorded, i.e. how was the instrument administered? Did people respond using pen & paper or electronic device? Did someone accompany or help them? Who did they return the instrument to, and was it anonymous? These considerations are critical when we consider what biases may have been at play during participation, and will also warrant reflection in the study limitations.

Line 155 consider stating that you tested for normality before using the t-test.

Methods, general. You describe in some detail the interventional study (psychologist consultation, telephone follow-up) but I was not clear until reading the Results that these were not to be covered again. You could greatly condense the description of the parent RCT and be clearer that in this paper you are going to be reporting the emotions sub-study.

Results, general. You mentioned open-ended questions for attendance reason and emotion - were these analysed?

Line 321 I don't think a sentence or citation on speed or epidemiology of traffic mortality are needed in this paper.

Discussion, general. This is a very thorough and well-referenced review, but comprehensiveness here comes at the expense of conciseness. The length makes the section difficult to digest and you could consider focusing on brevity.

Conclusions, general. There are many statements here containing conjecture or opinion. Suggest keeping to reporting a brief summary of facts supported by the presented results.

Author Response

Thank you for the invitation to review this interesting study exploring the emotions perceived by patients during ED attendance. I recommend some revisions, mostly minor, based on my comments below:

               Thanks for your comments

Comment 1: Line 15 'interviewed' - that word made me expect to see a qualitative study, whereas this was a survey / instrument administration with quantitative analysis. Suggest avoid 'interview'.

Response 1: We have modified the vocabulary for “surveyed”.

Comment 2: Line 62 typo, 'addiction'

Response 2: Thank you for reporting this error. We have added the missing character

Comment 3: Line 64-65 re no other study documenting emotions experienced by ED patients. I am aware of a growing literature base for ED patient experience (including emotions and perceptions) and its measurement:
https://doi.org/10.1016/j.ienj.2009.05.004
https://doi.org/10.1177/2374373517731359

(I advise the authors to conduct a literature review)

Response 3: The reviewer is right to point out the clumsiness of our sentence. Indeed, during our literature searches we focused on methodologies that allowed comparison with our study, and therefore much larger sample sizes than those of the qualitative studies cited in his reviews.

We have therefore modified the formulation in the introduction, lines 76-78:

“A handful of studies have documented the presence of emotions in EDs [20], but none have been carried out among study populations large enough to conduct any sort of quantitative analysis”

Comment 4: If you mean something by 'emotion' that is distinct from experience otherwise, suggest include an accessible definition in the Introduction.

Response 4: Indeed, the definition of emotions was missing from the article, and we have rectified this error. In addition, we have clarified the dimension of the feelings studied, which are reactive responses to restore homeostasis, rather than global appraisal during the emergency stay.

we have deleted this sentence and so added the following sentence in the introduction, lines 73-75:

“Emotions are defined by Scherer as “an episode of interrelated, synchronised changes in the states of all or most of the five organismic subsystems in response to the evaluation of an external or internal stimulus event as relevant to major concerns of the organism” [20]. A handful of studies have documented the presence of emotions in EDs [20], but none have been carried out among study populations large enough to conduct any sort of quantitative analysis.”

Comment 5: Line 97-98 re there being no other studies exploring feelings experienced in the ED. Again, I am not at all convinced by this statement as I am aware of a fairly substantial body of qualitative evidence stemming from ED enquiry. 

Response 5: In line with the previous comment, we correct this clumsy expression by referring to the quantification of emotional feelings.

We have modified the sentence, lines 111-112:

“Furthermore, to our knowledge, there is currently no other study that has quantified the emotions experienced in the ED”

Comment 6: Line 108 on. The enrolment period lasted 9 months, but recruitment was carried out over two weeks. Are you saying that the intervention / control alternated every week for 9 months, or did you recruit at each site for only two weeks during the 9 months? I was not clear but imagine the latter based on overall sample.

Response 6: We have added this clarification to the text, lines 122-127:

 “The study took place over 2 weeks in each centre. One week involved the intervention of a team of psychologists in addition to the usual care provided (intervention week), and during the other week care was provided as usual (control week). Randomisation was based on the order of intervention and control weeks. The same team of psychologists visited all 7 centres successively. The enrolment period for the study commenced on 18 October 2021 and lasted 9 months”

Comment 7: Line 110 please state explicitly whether e, or whether the 3132 patients assessed for eligibility relates to some other sampling frame e.g. the opportunistic availability of researchers etc. This is important to know so that readers can make some judgement as to the comprehensiveness of sampling (ie, is it possible that people with certain characteristics were not approached or assessed for eligibility).

Response 7: We take up your words to clarify the inclusion of patients and the absence of selection on criteria other than those of eligibility.

We have modified the methodology in accordance with the comments of reviewer 1 and 2, lines 129-130:

“SOFTER’s inclusion criteria were compatible with the feelings study, and every eligible patient was recruited consecutively following the ED admission.”

Comment 8: Line 121 how were feelings recorded, i.e. how was the instrument administered? Did people respond using pen & paper or electronic device? Did someone accompany or help them? Who did they return the instrument to, and was it anonymous? These considerations are critical when we consider what biases may have been at play during participation, and will also warrant reflection in the study limitations.

Response 8: Patients completed the questionnaire containing all the study variables on a tablet, in most cases by themselves. When necessary or when requested by the patients, they were given the option of a paper version of the questionnaire. The interviewer could also complete the questionnaire for patients who were physically unable to do so. In this case, the interviewers simply read out the instructions provided on the self-assessment questionnaire.

This clarification has been added to the lines 144-149:

This study is based on self-reported feelings. It is therefore possible that some patients did not report their feelings accurately, for example due to a lack of emotional awareness.

We add this clarification to the weakness paragraph of our discussion, lines 489-491:

            Comment 9: Line 155 consider stating that you tested for normality before using the t-test.

Response 9: We have added the precision, line 204.

Comment 10: Methods, general. You describe in some detail the interventional study (psychologist consultation, telephone follow-up) but I was not clear until reading the Results that these were not to be covered again. You could greatly condense the description of the parent RCT and be clearer that in this paper you are going to be reporting the emotions sub-study.

Response 10: We have largely modified the paragraph 2.1 of the methodology and deleted information not necessary to understand the results on feelings.

Comment 11: Results, general. You mentioned open-ended questions for attendance reason and emotion - were these analysed?

Response 11: The emotions collected in the open-ended question were used in the discussion to show that the Emotion Wheel used here had not missed any of the emotions frequently encountered in the emergency department. They were not described in detail in this study.

Comment 12: Line 321 I don't think a sentence or citation on speed or epidemiology of traffic mortality are needed in this paper.

Response 12: We have simplified the discussion by deleting the elements on the severity of falls and road accidents.

Comment 13: Discussion, general. This is a very thorough and well-referenced review, but comprehensiveness here comes at the expense of conciseness. The length makes the section difficult to digest and you could consider focusing on brevity.

Response 13: We have :

- removed items on the severity of falls and road accidents.

- simplified the coping aspects for women and older people

- removed the figures lines 397-398

Comment 14: Conclusions, general. There are many statements here containing conjecture or opinion. Suggest keeping to reporting a brief summary of facts supported by the presented results.

Response 14: We have followed the recommendations of reviewers 1 and 2 by proposing a conclusion more focused on a brief summary of facts and their clinical implications.

“The present study highlights the strong presence of pain, stress and emotions experienced by patients admitted to the ED. These feelings vary according to socio-demographic and health characteristics, and they are correlated, albeit weakly. The management of feelings in the ED is essentially focused on pain, which is of course essential. However, our results suggest the need to also manage also stress and emotions and encourage further work to define what types of management of these feelings could complement current protocols. It could also be useful to monitor these feelings over time, and asses how they are impacted by the context in which they are expressed.”

Reviewer 3 Report

Comments and Suggestions for Authors

1. The study is a well documented and written study.

2. Please provide more information on the measures: who is the author, whether they were used before, and where, etc.

3. Why this was necessary and why this way? "the number of years of study, divided into two catego- 148 ries (< 12 years, ≥ 12 years)," Explain the necessity.

4. "but their intensity was 133 not evaluated." justify this

5. You have some statements in the discussions section such as: "Pain was significantly higher among female patients under 60 admitted for injury 300 ", higher than whom? You need a comparison reference. Correct this in all instances in the article. 

6. IF you performed means comparison please refer to comparisons between means. x is higher than y. but not x is higher. 

"Just like in Patel et al., women report experiencing greater pain intensity, even after 301 excluding patients with a pain level of 0-3."

7. IN the introduction section you could include also results from studies focusing on the link between visiting emergency rooms and the psychological states. Maybe some theories, and some studies from the point of psychology. 

8. The sociodemographic characteristics could be better documented in the introduction part. Be specific. Were there other studies finding similar results?

9. Was the waiting time in the emergeny room considered? How this variable relates to the variables in the study? What means waiting time? is there a large crowd of people? How long it was?

10. Were there considered other variables related to social support of the patients? Were they alone when they came to ER? How are patients generally treated in such institutions? It may matter!

Author Response

  1. The study is a well documented and written study.

Resp 1: Thank you for this comment

  1. Please provide more information on the measures: who is the author, whether they were used before, and where, etc.

Resp 2: The following paragraphs have been added to the methodology:

Lines 151-156:

“The Initiative on Methods, Measurement, and Pain Assessment in Clinical Trials (IMMPACT) endorses self-assessment as the gold standard for pain measurement (Dworkin et al., 2005). In the case of people capable of self-assessment, the most commonly used tool is the numerical rating scale (NRS) (Williamson & Hoggart, 2005). Widely used in the medical field, this scale may have originally been developed by Dr. Melzack and Dr. Torgerson in 1975 at McGill University in the first Mc Gill pain questionnaire, with a version of the intensity scale ranging from 0 to 20.”

            Lines 156-159:

“For stress, a scale based on the NRS was recently developed in a paediatric clinical context by Karvounides et al. The aim was to provide an easy-to-use assessment scale with a short administration time to assess current stress.”

Lines 166-169:

“This is made possible by the Genova Emotion wheel, created by Scherer. As for pain and stress, it allows emotions to be measured in the moment, in a short format. It is mainly used in experimental studies requiring comparison between experimental groups.”

  1. Why this was necessary and why this way? "the number of years of study, divided into two

Resp 3: We've phrased it this way to be comparable with other education systems.

we have therefore added this precision to the method section, lines 192-193:

« since in France, the general secondary school diploma (Baccalauréat) is awarded after 12 years of schooling. “

  1. "but their intensity was 133 not evaluated." justify this

Resp 4: The emotions collected in the open-ended question were used in the discussion to show that the Emotion Wheel used here had not missed any of the emotions frequently encountered in the emergency department. They were not described in detail in this study.

It would have been interesting to also assess their intensity, but due to time constraints, this was not carried out.

  1. You have some statements in the discussions section such as: "Pain was significantly higher among female patients under 60 admitted for injury 300 ", higher than whom? You need a comparison reference. Correct this in all instances in the article. 

Resp 5: In the discussion, we repeated the results of the multivariate models, highlighting the category that cumulates the 3 significant associations: for pain, being a woman, being under 60, and being admitted for a traumatic reason. We implicitly compared them to those least associated with pain, men over 60 admitted for a medical reason.

In response to the comment, we have completed the reference categories in all the discussion.

For the rest of the discussion, we discussed the factors successively and modified the sub-headings.

Line 361:

Pain was significantly higher among female patients, patients under 60, and patients admitted for injury”

Line 393:

Stress was higher among female patients, patients under 60, and patients admitted for illness”

Line 438:

Negative emotions were higher among female patients, and patients admitted for injury. Positive emotions were higher among male patients, and patients over 60, and patients admitted for illness”

  1. IF you performed means comparison please refer to comparisons between means. x is higher than y. but not x is higher. 

Resp 6: In tables 3 and 4, we have compared the mean scores of feelings according to age, gender and reason for admission, using t-tests. We have not commented on these results in the results section. We have, however, added the reference classes to the discussion.

"Just like in Patel et al., women report experiencing greater pain intensity, even after 301 excluding patients with a pain level of 0-3."

In the discussion, as you suggest, we've added the comparison class when it wasn't specified.

  1. IN the introduction section you could include also results from studies focusing on the link between visiting emergency rooms and the psychological states. Maybe some theories, and some studies from the point of psychology. 

Resp 7: It is true that our study was based on an epidemiological approach. Nevertheless, as also suggested by reviewer 2, we added a reference to literature reviews based on qualitative methods that describe psychological states.

We have added this sentence to the introduction, lines 76-78:

“A handful of studies have documented the presence of emotions in EDs [20], but none have been carried out among study populations large enough to conduct any sort of quantitative analysis”

  1. The sociodemographic characteristics could be better documented in the introduction part. Be specific. Were there other studies finding similar results?

Resp 8: We selected studies that had a methodology comparable to ours, and found no other work that described pain, stress and emotional experiences, according to socio-demographic characteristics.

  1. Was the waiting time in the emergeny room considered? How this variable relates to the variables in the study? What means waiting time? is there a large crowd of people? How long it was?

Resp 9: As we suggested in the discussion, this is an interesting contextual element to consider in future work. Unfortunately, we only had this information for one of the 7 centers, and furthermore we could not reliably differentiate between waiting time for admission to the ward and waiting time for medical care.

We added the mention of the waiting time in the local context, line 502:

  1. Were there considered other variables related to social support of the patients? Were they alone when they came to ER? How are patients generally treated in such institutions? It may matter!

Resp 10: It's a very interesting piece of information indeed, but unfortunately it wasn't collected. We'll make a note of it for future work.

We added this mention to our discussion, lines 504-506:

“Feelings may have been influenced by the social support patients received during their time in the ED. However, this information was also not covered as part of our data collection.”

Round 2

Reviewer 1 Report

Comments and Suggestions for Authors

1.     Applicability of the Geneva Emotion Wheel: While the discussion includes a critical note on the Geneva Emotion Wheel, further elaboration on its applicability to populations outside France would enhance the manuscript. Specifically, the authors should address whether any cultural adjustments were made to the tool for this study, given its international usage and potential cultural variations in emotional expression.

2.     Positive Emotions in the Emergency Department: The manuscript briefly mentions the presence of strong positive emotions in patients upon entering the emergency department. However, the implications of these findings are underexplored. A more in-depth discussion of how such emotions might affect patient care, healthcare provider-patient interactions, or clinical outcomes could provide additional value and relevance to the study's findings.

Addressing these points will strengthen the manuscript further and enhance its contribution to the field.

Author Response

  1. Comment 1: Applicability of the Geneva Emotion Wheel: While the discussion includes a critical note on the Geneva Emotion Wheel, further elaboration on its applicability to populations outside France would enhance the manuscript. Specifically, the authors should address whether any cultural adjustments were made to the tool for this study, given its international usage and potential cultural variations in emotional expression.

Rep 1:

Some of the basic emotions commonly found on emotion wheels have been retained. Certain emotions have been selected as they were supposed to correspond to the French emergency context. We add this clarification.

Lies 470-479: “The Geneva emotion wheel requires a balance between the different dimensions, and thus for valency, the presence of the same number of positive and negative emotions. Apart from the primary emotions of fear, sadness and anger for the negative emotions, and joy for the positive emotions, which are initially present on the wheel, the emotions of regret, relief, satisfaction and interest were specifically included in the study's emotion wheel, as they were assumed to correspond to emotions that could be aroused by a health event for “regret”, or a care situation for the other positive emotions. It is therefore possible that they are not adapted to other contexts, such as other care situations or countries where common emotional expression differs from France.”

  1. Comment 2: Positive Emotions in the Emergency Department: The manuscript briefly mentions the presence of strong positive emotions in patients upon entering the emergency department. However, the implications of these findings are underexplored. A more in-depth discussion of how such emotions might affect patient care, healthcare provider-patient interactions, or clinical outcomes could provide additional value and relevance to the study's findings.

Rep 2 : Less emphasis had been placed on positive emotions, whereas we found these results quite interesting. A paragraph on the possible implications of their presence in the emergency context, and particularly of the emotions of relief and interest, as well as future work to be developed on the mid-term clinical consequences, has been added.

Lines 451-460: “ Their presence in an ED context may be a sign that previous negative emotions are giving way to positive ones. However, the dynamics of these emotions could not be observed in our study. Nonetheless, their presence and, in particular, feelings of relief and interest could be interesting clinical indicators. Interest could presage success in the therapeutic alliance with caregivers and better adherence to the care pathway. As for appeasement, its presence is reassuring for both patients and health professionals. It may be the first step towards the resolution of the health problem that brought the patient to the service. Evaluating the association between the presence of positive emotions and recovery would provide further insight into their clinical relevance to health outcomes following a visit to the ED.”

Comment 3: Addressing these points will strengthen the manuscript further and enhance its contribution to the field.

Rep 3: Thanks to the reviewer for all his/her comments, which helped us to enrich the paper. We hope that these responses will enable us to complete it satisfactorily.

Reviewer 2 Report

Comments and Suggestions for Authors

The authors have provided a comprehensive response and addressed my comments. I have no further concerns or feedback and congratulate the authors on this endeavour.

Author Response

Comment: The authors have provided a comprehensive response and addressed my comments. I have no further concerns or feedback and congratulate the authors on this endeavour.

Response: Thanks to the reviewer for all his/her comments, which enriched the paper and his/her positive feedback on our work.

Reviewer 3 Report

Comments and Suggestions for Authors

Thank you for answering to all!

Author Response

Comment: Thank you for answering to all!

Response: Thanks to the reviewer for all his/her comments, which enriched the paper and his/her positive feedback on our work.